# Analysis and optimization of a Stochastic Petri Net for air-rail intermodal transportation

Yihu Lei*, Haibo Mu

School of Traffic and Transportation, Lanzhou Jiaotong University, Lanzhou City, Gansu Province, People's Republic of China

* rayhoblade@gmail.com, 11220048@stu.lzjt.edu.cn

## Abstract

Air-rail intermodal transportation (ARIT) plays a crucial role in China's intermodal transportation system. This study aims to model and optimize issues such as inefficiency and complexity in China's ARIT freight transportation using Business Process Reengineering and Stochastic Petri Nets theories. The Petri Net (PN) model for incoming freight transport in ARIT is based on actual operations, employing a new method involving Stochastic Petri Nets and isomorphic Markov Chains theory for performance analysis. Performance analysis results help intuitively identify areas needing optimization. Based on optimization principles, elements such as railway container packaging are improved, resulting in an optimized PN model for ARIT. Finally, data analysis shows that the optimized ARIT model reduces total delay by 7.7% compared to the original. This demonstrates that the new method, combining Markov Chain performance analysis and optimization principles, is feasible and effective for ARIT optimization.

**Data Availability Statement:** The research data for this study originates from the academic paper titled "Design and Simulation Study on Air-Rail Intermodal Transport Process at Zhengzhou Airport," with the reference formatted as: Ying

## Introduction

Significant progress has been made in the development of global Intermodal transportation all around the world, especially in Europe and China. Europe has gained extensive experience from early research and practice, significantly enhancing operational efficiency and passenger convenience through system optimization and intermodal cooperation [1–4]. Intermodal transportation has also contributed significantly to China's transportation system in recent years and has garnered considerable government attention. The "Work Plan to Promote the Development and Optimization of Intermodal Transport and Adjusting the Transport Structure," issued by the General Office of the State Council on December 25, 2021, emphasized the necessity of innovating organizational models in intermodal transport and exploring new intermodal transport modes to enhance China's transportation system. ARIT, which integrates aviation and railways, emerges as a crucial component within intermodal transport systems, which has started to be implemented at various airports in China, with its freight transport procedures gradually standardizing. The widespread adoption of air-rail intermodal freight transportation (ARIFT) contributes positively to the optimization of China's aviation and high-speed rail networks, the acceleration of overall freight efficiency, and the diversification of freight transport methods.

Feilong. Design and Simulation Study on Air-Rail Intermodal Transport Process at Zhengzhou Airport. Civil Aviation University of China, 2020. doi:10.27627/d.cnki.gzmhy.2020.000287."

**Funding:** The author(s) received no specific funding for this work.

**Competing interests:** The authors have declared that no competing interests exist.

In the realm of intermodal transport, many experts and scholars have extensively researched two main areas: optimizing intermodal transport path networks and refining business processes. Within the domain of optimizing path networks, Wu et al. [5] developed a multi-objective optimization model for intermodal transport, focusing on minimizing carbon emissions, transportation costs, and time across various transportation modes. They validated the effectiveness by applying it to cargo transportation via the China-Europe Railway Express. Gao [6] developed a hybrid objective optimization model aimed at minimizing total freight costs and time, enhancing intermodal transport freight path optimization using the DQN algorithm. The practical applicability of the improved DQN algorithm was validated. Hu [7] et al. tackled the optimization challenge of intermodal transport paths, accounting for constraints like time windows, fixed departure times, and delivery windows. They applied a robust optimization approach and validated it with genetic algorithms, finding that alterations in node time windows impacted costs and solutions. Duan [8] et al. investigated the optimization of cold chain intermodal transport paths using an enhanced immune optimization algorithm. They characterized cold chain losses with a Weibull three-parameter distribution, devised a model to minimize overall transportation expenses, and validated it with particle swarm optimization algorithms. Yuan et al. [9] introduced a two-stage model to optimize freight pricing and transportation routes for intermodal container rail-road transport. Numerical analysis indicated that the profits of small-scale transporters were approximately 11.42% higher than those of large-scale transporters. Wang et al. [10] customized a subway train operation plan for comprehensive passenger hubs, developing a dual-objective integer nonlinear programming model. They obtained Pareto optimal solutions using real-time traffic prediction methods and variable neighborhood search algorithms, validating their efficacy through numerical experiments. Song et al. [11] investigated route selection for transporting hazardous goods in railroad intermodal transport, employing a bi-objective optimization approach, and considering alternative routes when highways are temporarily inaccessible. Case studies demonstrated that the use of alternate routes could improve overall transport efficiency.

In the realm of optimizing business processes, Liu [12] employed Petri nets (PN) to develop a cargo transportation process model for ARIT. They simulated the air-rail intermodal cargo transportation processes both pre and post-optimization using Flexsim simulation software, demonstrating the optimization's effectiveness through simulation results. Ying [13] systematically examined the cargo transport process within Zhengzhou Airport ARIT, employing a comprehensive approach that integrates PN modeling and Flexsim simulation. They suggested optimization measures and verified a 27.8% efficiency improvement in the optimized process, offering practical insight for China's airport cargo transportation. Xiang [14] devised a public-rail intermodal information flow process employing an information platform. They identified collaboration points for information sharing using PN and introduced new shared information attributes to improve economic benefits and reduce intermodal transportation times, illustrated with general cargo as an example. Li [15] investigated the effect of conflicting information on the efficiency of public-rail intermodal transport business processes, considering both the customer and the manager viewpoints. They developed overall process network efficiency models and colored time business process network efficiency models from both perspectives. The results indicated a higher operational efficiency from the customer point of view, offering optimization strategies for public-rail intermodal transport business processes. Wang et al. [16] constructed and simulated intermodal transport business processes using extended PN, identifying system conflicts and suggesting optimization remedies. They conducted simulations to validate the feasibility of optimization solutions. Mariagrazia et al. [17] examined the efficiency of intermodal transport systems using timed PN theory, validating its efficacy through case simulations and offering additional transport solutions for intermodal

terminal transport. Graziana et al. [18] developed a model for intermodal transport process planning and resource management, integrating timed Petri nets(TPN) and discrete data analysis. They validated the model with practical cases, highlighting the suitability of discrete data analysis for transport decision making amid demand conflicts.

Stochastic Petri nets (SPN) constitute a sophisticated mathematical network optimization model capable of constructing Markov chains (MC) for process networks and conducting optimization analyzes. Within the realms of SPN and MC, Liao [19] concentrated on the inbound and outbound dual-mode operations' business processes at Chengdu Tianfu Airport. Integrating the airport's current operational status with SPN, they optimized certain aspects of the model and employed MC to assess process performance pre and post-optimization. They validated the model optimization viability by computing the incurred delay time. Zhang and colleagues [20] examined issues within the railway freight business process under the e-commerce platform employing SPN theory. They adopted a hierarchical modeling approach to pinpoint irrational structures in the model, substantially enhancing freight process efficiency post-improvement. Yang and colleagues [21] scrutinized the export operation process at Zhengzhou Railway Container Center Station utilizing SPN model. They proposed multilevel optimization strategies to enhance operational efficiency and service quality at the China-Europe Railway Express station. Danial and team [22] developed a novel environmental route learning model grounded in generalized SPN, mimicking human behavior to identify the most efficient path for decision-makers amidst stochastic conditions. Wang and colleagues [23] characterized the work content of emergency departments utilizing SPN theory, assessing the effect of personnel configuration on service performance and optimizing the emergency department's personnel layout accordingly. Zhao and colleagues [24] examined safety system integrity levels employing a blend of SPN and Monte Carlo simulation theory. They contrasted the optimized application model with the standard reliability block diagram, affirming the viability of integrating SPN with Monte Carlo simulation techniques. Laura and team [25] developed a customer selection model grounded in MC selection models and reservation price theory, deriving optimal solutions in discrete scenarios to tackle intricate issues like customer selection. Sun and colleagues [26] employed SPN theory in responding to the COVID-19 pandemic emergency, using Xi'an City as a case study to formulate a model for sudden disease outbreak responses. They utilized rescue time uncertainty to establish MC for emergency model development, identifying critical positions in emergency coordination based on calculated probabilities, offering pertinent recommendations for emergency event prevention.

In summary, research on intermodal transport has primarily concentrated on optimizing path networks, with a majority of studies focusing on road-rail intermodal transport. However, research on optimizing ARIFT operation processes is still nascent. Certain domestic literature has explored the business processes of ARTFT, utilizing PN theory to optimize and simulate constructed models. Building on the aforementioned literature, this paper presents a novel approach employing SPN to examine air-rail intermodal freight operation processes. Drawing from the theory of SPN, which is isomorphic to MC, MC can be derived from the PN model. Stable-state probabilities derived from the MC can more intuitively pinpoint areas that require optimization within processes. Additionally, they facilitate the calculation of delays before and after system optimization, showcasing the effectiveness of optimization measures. MC theory enables mathematical analysis of the incoming freight process, offering more logical mathematical proofs and theoretical backing for analyzing and optimizing air-rail intermodal incoming freight operation processes, and proposing enhancements for domestic air-rail intermodal freight operation processes. This study focuses on domestic airports that provide air-rail intermodal freight services, establish inbound freight process models for ARIT, optimize

them, and utilizing delay data from air-rail intermodal transport operations at Zhengzhou Airport to perform a performance analysis of inbound freight processes employing MC theory.

## Materials and methods

### Theoretical framework

**Stochastic petri net theory.** Stochastic Petri Nets (SPN) extend traditional Petri net theory to model and analyze discrete event dynamic systems. Originally proposed by Carl Adam Petri in 1962, Petri nets are a mathematical model for describing and analyzing concurrent systems. SPNs extend this by associating random variables with transition firing times, imparting stochastic behavior to the system. This approach enables SPNs to simulate random phenomena in complex systems and provides tools for analyzing system performance and reliability. The main types of SPNs include Generalized Stochastic Petri Nets (GSPN), Extended Stochastic Petri Nets (ESPN), and Deterministic and Stochastic Petri Nets (DSPN). Each type introduces different stochastic elements and structures to meet the modeling needs of various systems [27,28].

SPNs are widely used in performance evaluation, reliability analysis, communication systems, and manufacturing systems. For example, they analyze the performance of multiprocessor systems, the reliability of network protocols, and the behavior of complex systems. By simulating and analyzing random events within these systems, researchers can better understand their dynamic behavior and performance bottlenecks. The main analysis methods for SPNs include state space analysis, simulation, and Markov chain methods. State space analysis enumerates all possible states and transitions, providing a detailed description of system behavior. Simulation methods estimate performance metrics by running multiple simulation instances. Markov chain methods model the system's stochastic behavior as a Markov process for quantitative analysis. The primary advantage of SPNs is their ability to accurately describe stochastic behavior in concurrent systems and offer powerful analytical tools. However, they can be complex and computationally intensive, especially with large state spaces, where analysis can become intricate and time-consuming.

**Markov chain theory.** The Markov Chain (MC) is a mathematical model for describing stochastic processes, where the future state depends only on the current state, not on past states. This characteristic is known as the Markov property. This "memorylessness" makes MCs widely applicable in various fields. MCs can be classified into Discrete-Time Markov Chains (DTMC) and Continuous-Time Markov Chains (CTMC). In DTMC, states transition at discrete time points, whereas in CTMC, states transition over continuous time [29].

Methods for analyzing MCs mainly include state space analysis and transition probability matrix analysis. State space analysis involves calculating all possible states and their transition probabilities. Transition matrix analysis predicts the state distribution at future time points through repeated matrix multiplications. The main advantages of MCs are their simple mathematical model and computational convenience, making them suitable for various stochastic processes. However, their disadvantages include potentially high computational complexity for systems with large state spaces and the need for extensive data to accurately estimate transition probabilities in practice.

### Modeling and analysis of intermodal freight transport process in ARIT

Modeling the intermodal freight transport process for ARIT involves the flow of goods through airports, air-rail intermodal centers, and high-speed rail stations, where a freight flow chart is constructed, modeled, and optimized. Business Process Reengineering (BPR) theory is employed in incoming freight transportation for intermodal air-rail transportation. BPR

theory is a business management strategy focused on analyzing and redesigning workflow and processes within an organization to enhance customer service, cut operational costs, and facilitate quicker adaptation to market changes. In intermodal freight transport for ARIT, the necessity to transport goods between major cities underscores the importance of ensuring efficient goods transportation, given the immediacy of air transport and the timeliness of high-speed rail freight transport. The transportation duration via air routes and high-speed rail networks largely determines the efficiency of ARIFT. Therefore, optimizing the efficiency of goods circulation in air-rail intermodal centers is critical to improve the overall efficiency of ARIFT and to take advantage of the timeliness advantage inherent in this mode of transport. The research direction predominantly centers on the ARIFT center connecting air and railway transport, encompassing activities such as loading and unloading, sorting and packaging, and weighing of goods within the air-rail intermodal freight transportation center. Therefore, optimizing incoming freight transportation processes for ARIT begins with activities within the ARIFT center, traverses various functions and departments, comprehensively evaluates, analyzes, and optimizes business processes from an overarching optimization perspective, improving service efficiency in ARIFT, reducing transit times, and meeting customer needs.

**Diagram of incoming freight transport operations for ARIT.** The incoming freight transport operations process for ARIT involves multiple departments and progresses sequentially through three primary stages: apron, air-rail intermodal center, and high-speed rail station. This process entails transporting goods from the aircraft's landing at the airport to the cargo station post-order verification, followed by sequential operations including sorting, weighing, document handover, loading, and the handling of air cargo container disassembly and railway container packaging, culminating in the departure of the high-speed rail from the station. Additionally, it involves the exchange of information and funds flows between departments. Fig 1 illustrates the incoming freight transport operations process for ARIT, where each process link operates independently and influences only the subsequent link in progression.

Fig 1 illustrates the operational process of ARIT's incoming freight transport, based on reference [13], integrating the processes of Zhengzhou Xinzheng International Airport and Zhengzhou South Station. The first row of the figure encompasses the three main stages of ARIT's incoming freight operations: apron, air-rail intermodal center, and high-speed rail station. The process begins with the aircraft landing, followed by reception and order verification by freight personnel. Subsequently, the cargo is offloaded at the apron and transported by external trailers. At the air-rail intermodal center, the cargo undergoes disassembly, reassembly, weighing, and packaging into railway containers, with the final step being document exchange. In the high-speed rail station stage, freight personnel devise and organize the loading plan, conduct pre-departure inspections, and finally, the high-speed train departs and enters the railway network, completing the entire freight operation process. Each stage operates independently, directly influencing the next. The PN of ARIT model constructed in this paper is based on the described process, ensuring alignment with practical application. This provides a solid foundation for performance analysis and model optimization.

**PN model of ARIFT incoming operations process.** This paper employs PN theory to hierarchically model the incoming freight operation process of ARIT based on the described freight operation process. The modeling design of the ARIFT process in this paper is based on several assumptions: (1) Goods are primarily transported as ordinary freight, with high-speed rail vehicles constituting the entire freight train. (2) Given the exploratory nature of the ARIFT process, the design in this document focuses solely on domestic cargo intermodal transport. (3) Each process is assumed to proceed smoothly without any special circumstances. (4) It is assumed that goods transported by high-speed rail freight trains upon entry are not local and

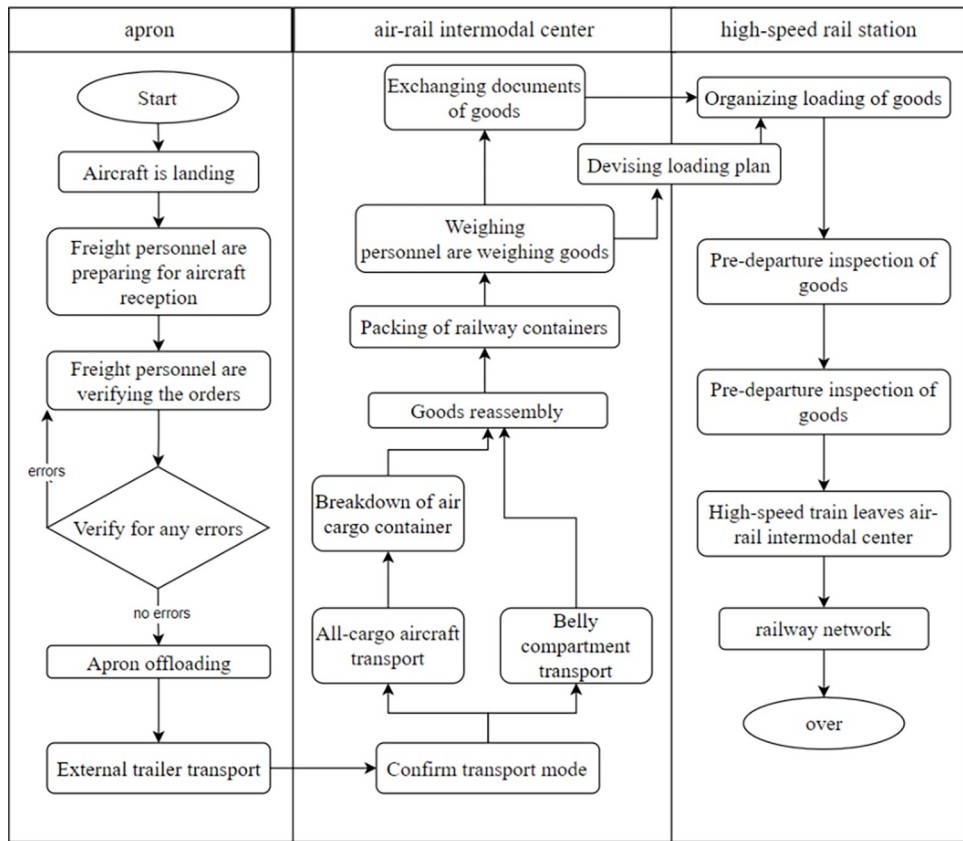

**Fig 1. Flowchart.**

all require air freight transportation upon leaving the port; conversely, goods arriving via air freight do not include local goods and all require transportation by high-speed rail freight trains. This paper selects the incoming freight operation process of domestic ARIT for PN modeling and research after fulfilling the aforementioned assumptions. This process encompasses aircraft arrival at the airport, goods delivery to the ARIFT center, sorting and assembly operations, loading onto high-speed rail trains, and integration into the main railway network. This article constructs the PN model depicting the incoming freight operation processes of domestic ARIT, as illustrated in Fig 2, according to the characteristics of PN. The meanings of the places and transitions in the model are detailed in Table 1. Fig 2 contains 27 places and 24 transitions. In the PN model, circles represent places, squares represent transitions, and arrows connect places and transitions. Places cannot be directly connected to other places, and transitions cannot be directly connected to other transitions, as this violates the basic principles of PN.

The PN model depicted in Fig 2 contains precisely one token in p1, signifying its restriction to analyzing a singular set of incoming freight operation process within ARIT. The specific description of the process model for incoming freight operations in domestic ARIT is as follows: p1 symbolizes the aircraft's arrival, preparing for landing, initiating transition "Aircraft Landing" (t1), and utilizing one token to p2 (aircraft completes landing and readies for apron entry). Subsequently, place p2 activates transition "Aircraft Entering Apron" (t2), expending one token to advance to p3 (freight personnel readiness for entry). Following this, p3 initiates transition "Freight Personnel Entering" (t3), utilizing one token to proceed to p4. Place p4

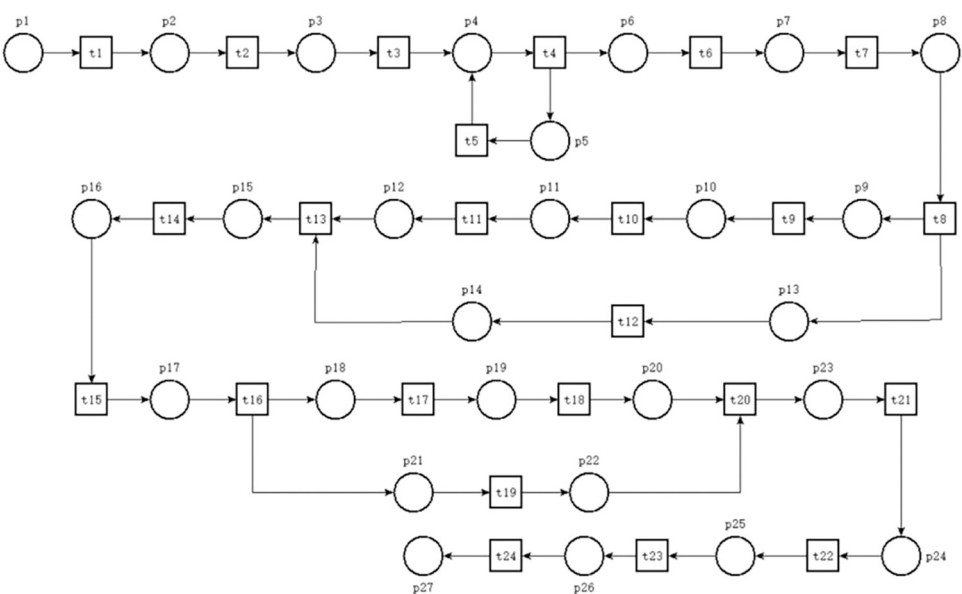

**Fig 2. The PN of domestic ARIFT incoming operation process.**

**Table 1. Meanings of places and transitions in PN of domestic ARIFT incoming operations.**

| Place (p) | Meanings | Transition(t) | Meanings |
|---|---|---|---|
| p1 | Aircraft arrives at the airport, preparing to land | t1 | Aircraft is landing |
| p2 | Aircraft completes landing, entering apron | t2 | Aircraft is entering apron |
| p3 | Freight personnel prepare to enter | t3 | Freight personnel are entering |
| p4 | Freight personnel prepare to verify orders | t4 | Freight personnel are verifying orders |
| p5 | Error during order verification | t5 | Processing erroneous orders |
| p6 | Orders verified, freight personnel prepare unloading plan | t6 | Freight personnel are devising unloading plan |
| p7 | Unloading plan finished, freight personnel prepare to unload | t7 | Freight personnel are unloading |
| p8 | Unloading completed, prepare for external trailer transport | t8 | External trailer transport |
| p9 | Goods arrive at cargo handling area, preparing to confirm transport mode | t9 | Confirming transport mode |
| p10 | Confirmed as all-cargo aircraft transport | t10 | Goods are undergoing all-cargo aircraft transport |
| p11 | Prepare for breakdown of air cargo container | t11 | Breakdown of air cargo container |
| p12 | Breakdown completed, goods ready for transport to sorting area | t12 | Goods are undergoing belly compartment transport |
| p13 | Confirmed as belly compartment transport | t13 | Goods are entering sorting area |
| p14 | Goods prepare to enter sorting area | t14 | Goods reassembly |
| p15 | Sorting personnel prepare goods reassembly | t15 | Packing of railway containers |
| p16 | Reassembly completed, prepare packing of railway containers | t16 | Goods being transported to weighing area |
| p17 | Packing of railway containers completed, goods ready for weighing area transport | t17 | Weighing personnel are weighing goods |
| p18 | Weighing personnel prepare to weigh goods | t18 | Goods are undergoing external trailer transport |
| p19 | Weighing completed, prepare for external trailer transport | t19 | Loading personnel are devising loading plan |
| p20 | Goods weighed, prepare for loading | t20 | Organizing loading of goods |
| p21 | Loading personnel prepare for loading plan | t21 | Inspecting after goods loaded |
| p22 | Loading plan prepared, goods ready for loading | t22 | Exchanging documents of goods |
| p23 | Loading completed, prepare for post-loading inspection | t23 | Pre-departure inspection of goods |
| p24 | Goods ready for document exchange | t24 | High-speed train is departing air-rail intermodal freight center |
| p25 | Goods are prepared for post-loading inspection | | |
| p26 | High-speed train prepares to depart | | |
| p27 | High-speed train leaves air-rail intermodal center, enters railway network | | |

signifies readiness for order status verification, initiating the transition "Freight Personnel Verifying Orders" (t4). Following the token consumption, it enters a loop. In case of accurate verification, the token depletes, shifting to p6 through t4; in case of erroneous orders, the token depletes, engaging a loop with p5 and t5 for incorrect order management. Upon completion, it returns to p4 for continued verification until all orders are correct, terminating the loop. Place p6 indicates completion of order verification, prompting freight personnel to formulate unloading plans, activating transition "Develop Unloading Plan" (t6), and utilizing one token to p7 (freight personnel readiness for unloading). Subsequently, t7 is started (freight personnel unloading), while place p8 (status of goods transported by external trailer) acquires the token. Following t8 (external trailer load of goods), a parallel relationship is established, signifying concurrent transport via full freighter and belly cargo transport. Upon confirmation of the goods transport mode, p10 and p13 "Full Freight Transport" and "Belly Cargo Transport" are generated. Full freight transport initiates t10 (goods transported via full freighter), traversing p11 (status of disassembled air container) and activating t11 (disassembly of air container) to p12 (goods prepared for transport to sorting area). During the same time, the loading of the cargo to the belly triggers t13 (goods entering sorting area), leading to p14 (status of goods transported to the sorting area). Place p12 or p14 initiates the "Goods Transported to Sorting Area" transition (t13), enabling p15 (status of goods ready for assembly) to acquire one token. Following this, p15 activates the transition "Goods Assembly" (t14), proceeding to p16 (preparation for packaging into railway container), while p17 (goods readiness for weighing) receives one token post-depletion. Transition t16 (weighing process) initiates a parallel structure, encompassing p18 and p21 denoting "Preparing for Weighing" and "Preparing for Loading Plan", respectively. Place p18 activates t17 (goods weighing), progressing to place p19 (preparation for external trailer transport) and further activating t18 (external trailer transport of goods) to advance to p20 (goods ready for loading). Concurrently, p21 initiates the transition "Develop Loading Plan" (t21), leading to p22 (good readiness for loading). Subsequently, p20 and p22 collectively activate the "Goods Loading" transition(t20), facilitating p23 (status of goods readiness for pre-loading inspection) to acquire new token. Transition t21 (inspection of goods loading) leads to p24 (preparation for document handover), activating t22 (document handover) prior to p24 token depletion and to p25 (pre-departure inspection preparation), enabling transition t23 (pre-departure inspection). Place p26 (high-speed rail readiness to depart) initiates t24 (high-speed rail departure from air-rail intermodal center), spending one token to enable p27 (train entering rail network) to acquire new token. At this juncture, the entire inbound freight operation process of ARIT concludes.

**Analysis of the PN model for ARIFT incoming operations processes.** The analysis of the ARIFT incoming operations process PN model, utilizing the adjacency matrix method, exposes the model's input and output matrices. Subtracting the input matrix from the output matrix yields the PN model of the adjacency matrix A of the ARIFT incoming operations process, depicted in Fig 3.

The formula $A^{T}X = 0$ allows for the determination of the invariants of S_X Concurrently, a basis for the solution set X can be derived:

$$X_1 = (1, 1, 1, 1, 0, 0, 0, 0, 0, 0, 0, 0, 0, 0, 0, 0, 0, 0, 0, 0, 0, 0, 0, 0, 0, 0, 0)^{T}$$

$$X_2 = (0, 0, 0, 0, 0, 0, 0, 0, -1, -1, -1, -1, 1, 1, 0, 0, 0, 0, 0, 0, 0, 0, 0, 0, 0, 0, 0)^{T}$$

$$X_3 = (0, 0, 0, 0, 0, 0, 0, 0, 0, 0, 0, 0, 0, 0, 0, 0, 0, -1, -1, -1, 1, 1, 0, 0, 0, 0, 0)^{T}$$

$$A=\begin{pmatrix}
-1 & 0 & 0 & 0 & 0 & 0 & 0 & 0 & 0 & 0 & 0 & 0 & 0 & 0 & 0 & 0 & 0 & 0 & 0 & 0 & 0 & 0 & 0 & 0 \\
1 & -1 & 0 & 0 & 0 & 0 & 0 & 0 & 0 & 0 & 0 & 0 & 0 & 0 & 0 & 0 & 0 & 0 & 0 & 0 & 0 & 0 & 0 & 0 \\
0 & 1 & -1 & 0 & 0 & 0 & 0 & 0 & 0 & 0 & 0 & 0 & 0 & 0 & 0 & 0 & 0 & 0 & 0 & 0 & 0 & 0 & 0 & 0 \\
0 & 0 & 1 & -1 & 1 & 0 & 0 & 0 & 0 & 0 & 0 & 0 & 0 & 0 & 0 & 0 & 0 & 0 & 0 & 0 & 0 & 0 & 0 & 0 \\
0 & 0 & 0 & 1 & -1 & 0 & 0 & 0 & 0 & 0 & 0 & 0 & 0 & 0 & 0 & 0 & 0 & 0 & 0 & 0 & 0 & 0 & 0 & 0 \\
0 & 0 & 0 & 1 & 0 & -1 & 0 & 0 & 0 & 0 & 0 & 0 & 0 & 0 & 0 & 0 & 0 & 0 & 0 & 0 & 0 & 0 & 0 & 0 \\
0 & 0 & 0 & 0 & 0 & 1 & -1 & 0 & 0 & 0 & 0 & 0 & 0 & 0 & 0 & 0 & 0 & 0 & 0 & 0 & 0 & 0 & 0 & 0 \\
0 & 0 & 0 & 0 & 0 & 0 & 1 & -1 & 0 & 0 & 0 & 0 & 0 & 0 & 0 & 0 & 0 & 0 & 0 & 0 & 0 & 0 & 0 & 0 \\
0 & 0 & 0 & 0 & 0 & 0 & 0 & 1 & -1 & 0 & 0 & 0 & 0 & 0 & 0 & 0 & 0 & 0 & 0 & 0 & 0 & 0 & 0 & 0 \\
0 & 0 & 0 & 0 & 0 & 0 & 0 & 0 & 1 & -1 & 0 & 0 & 0 & 0 & 0 & 0 & 0 & 0 & 0 & 0 & 0 & 0 & 0 & 0 \\
0 & 0 & 0 & 0 & 0 & 0 & 0 & 0 & 0 & 1 & -1 & 0 & 0 & 0 & 0 & 0 & 0 & 0 & 0 & 0 & 0 & 0 & 0 & 0 \\
0 & 0 & 0 & 0 & 0 & 0 & 0 & 0 & 0 & 0 & 1 & 0 & -1 & 0 & 0 & 0 & 0 & 0 & 0 & 0 & 0 & 0 & 0 & 0 \\
0 & 0 & 0 & 0 & 0 & 0 & 0 & 1 & 0 & 0 & 0 & -1 & 0 & 0 & 0 & 0 & 0 & 0 & 0 & 0 & 0 & 0 & 0 & 0 \\
0 & 0 & 0 & 0 & 0 & 0 & 0 & 0 & 0 & 0 & 0 & 1 & -1 & 0 & 0 & 0 & 0 & 0 & 0 & 0 & 0 & 0 & 0 & 0 \\
0 & 0 & 0 & 0 & 0 & 0 & 0 & 0 & 0 & 0 & 0 & 0 & 1 & -1 & 0 & 0 & 0 & 0 & 0 & 0 & 0 & 0 & 0 & 0 \\
0 & 0 & 0 & 0 & 0 & 0 & 0 & 0 & 0 & 0 & 0 & 0 & 0 & 1 & -1 & 0 & 0 & 0 & 0 & 0 & 0 & 0 & 0 & 0 \\
0 & 0 & 0 & 0 & 0 & 0 & 0 & 0 & 0 & 0 & 0 & 0 & 0 & 0 & 1 & -1 & 0 & 0 & 0 & 0 & 0 & 0 & 0 & 0 \\
0 & 0 & 0 & 0 & 0 & 0 & 0 & 0 & 0 & 0 & 0 & 0 & 0 & 0 & 0 & 1 & -1 & 0 & 0 & 0 & 0 & 0 & 0 & 0 \\
0 & 0 & 0 & 0 & 0 & 0 & 0 & 0 & 0 & 0 & 0 & 0 & 0 & 0 & 0 & 0 & 1 & -1 & 0 & 0 & 0 & 0 & 0 & 0 \\
0 & 0 & 0 & 0 & 0 & 0 & 0 & 0 & 0 & 0 & 0 & 0 & 0 & 0 & 0 & 0 & 0 & 1 & 0 & -1 & 0 & 0 & 0 & 0 \\
0 & 0 & 0 & 0 & 0 & 0 & 0 & 0 & 0 & 0 & 0 & 0 & 0 & 0 & 0 & 0 & 0 & 1 & 0 & 0 & -1 & 0 & 0 & 0 \\
0 & 0 & 0 & 0 & 0 & 0 & 0 & 0 & 0 & 0 & 0 & 0 & 0 & 0 & 0 & 0 & 0 & 0 & 1 & -1 & 0 & 0 & 0 & 0 \\
0 & 0 & 0 & 0 & 0 & 0 & 0 & 0 & 0 & 0 & 0 & 0 & 0 & 0 & 0 & 0 & 0 & 0 & 0 & 1 & -1 & 0 & 0 & 0 \\
0 & 0 & 0 & 0 & 0 & 0 & 0 & 0 & 0 & 0 & 0 & 0 & 0 & 0 & 0 & 0 & 0 & 0 & 0 & 0 & 1 & -1 & 0 & 0 \\
0 & 0 & 0 & 0 & 0 & 0 & 0 & 0 & 0 & 0 & 0 & 0 & 0 & 0 & 0 & 0 & 0 & 0 & 0 & 0 & 0 & 1 & -1 & 0 \\
0 & 0 & 0 & 0 & 0 & 0 & 0 & 0 & 0 & 0 & 0 & 0 & 0 & 0 & 0 & 0 & 0 & 0 & 0 & 0 & 0 & 0 & 1 & -1 \\
0 & 0 & 0 & 0 & 0 & 0 & 0 & 0 & 0 & 0 & 0 & 0 & 0 & 0 & 0 & 0 & 0 & 0 & 0 & 0 & 0 & 0 & 0 & 1
\end{pmatrix}$$

**Fig 3. The adjacency matrix A for the ARIFT incoming operations process of PN model.**

The invariant of the place set S_X is represented as S_X = $k_1X_1+k_2X_2+k_3X$. Since the solutions in the basis set $X_1$, $X_2$, $X_3$ are all non-negative, setting $k_1 = 1$ and $k_2 = k_3 = 0$ eliminates negative solutions from the invariant S_X, resulting in a logically consistent minimal invariant S_X. By combining properties such as PN liveness, reachability, and boundedness, we can conclude that the PN model of domestic ARIFT operation process is effective and feasible. The PN comprises four structural relationships: selection, conflict, synchronization, and concurrency. Among them, selection and concurrency relationships do not affect the network structure, thus not impacting the efficiency of the model. However, the model efficiency is affected by synchronization and conflict relationships. Therefore, as much as possible, converting synchronization and conflict relationships into selection and concurrency relationships is necessary. Based on the adjacency matrix A and the properties of these four structural relationships, the following phenomena are observed: The fourth row of the adjacency matrix A contains an additional 1, indicating the involvement of p4 in the selection relationship. In columns 4, 8, and 16 of the matrix, there are additional 1s, indicating the involvement of t4, t8, and t16 in the concurrency relationship. In columns 13 and 20 of the matrix, there are additional -1s, indicating the involvement of t13 and t20 in the synchronization relationship. Since there are no additional -1s in any rows of the adjacency matrix A, it is inferred that there are no conflict relationships in the model. In summary, the selection, conflict, synchronization, and concurrency relationships of the PN model depicted in Fig 2 are summarized in Table 2.

## Performance analysis and optimization for ARIFT incoming operations processes

To transform the general PN model depicted in Fig 2 into SPN model, time parameters conforming to a specific probability distribution must be incorporated into the transitions of the original model. These time parameters typically adhere to an exponential distribution and all transitions in the model are non-instantaneous. The time parameters for transitions are

**Table 2. Selection, synchronization, concurrency, conflict relationships in the PN model.**

| Selection | Synchronization | Concurrency | Conflict |
|---|---|---|---|
| p4 | t13,t20 | t4,t8,t16 | —— |

derived from delay data cited in reference [13]. Given that the data in this reference are presented in intervals, the median of each interval for each phase is utilized to assign values to transitions in the model, thereby acquiring the delay data for each transition.

**Performance analysis in for ARIFT incoming operations processes.** The PN model illustrated in Fig 2 encompasses 24 transitions, each characterized by time delays. Data from the literature provides delay information for these 24 transitions in Table 3, while Table 4 presents the implementation rates of transitions. Each transition conforms to an exponential distribution, enabling the transformation of the PN model into SPN model. Utilizing the SPN, reachable markings are determined, elucidating the corresponding places for each marking, as depicted in Table 5. Leveraging the reachable markings and rate assignment to arcs, the certain MC is established, which is isomorphic to the SPN, as delineated in Fig 4.

From the MC model of Incoming Operation Process in ARIFT, the transition rate matrix Q for the Markov process can be derived. Let $X = (x_1, x_2, x_3. \ldots . . x_{29}, x_{30})$ denote the probabilities of all markings in the MC. This is determined by the equation:

$$\begin{cases} XQ = 0 \\ \sum_{i=1}^{n} xi = 1, 1 \leq i \leq 29 \end{cases} \quad (1)$$

The probabilities of all markings can be acquired as presented in Table 6. This is achieved by considering the correlation between places and markings delineated in Table 5, along with the equation for computing the place occupancy rate:

$$P[M(s) = i] = \sum_{j} P[M_j] \quad (2)$$

The occupancy rate of places in the model can be calculated by Eq (2). For places corresponding to a single marking, the occupancy rate equals the steady-state probability of that marking. For places associated with multiple markings, the occupancy rate is the summation of the steady-state probabilities of those markings. The occupancy rates of places in the model are presented in Table 7.

The average token count of places in the model is found to be

$$\bar{N} = \sum_{i=1}^{27} u_i = \sum_{i=1}^{27} [p(Mi) = 1] \quad (3)$$

= 1.2323. The flow rate of incoming tokens into the system is $\lambda = u_1 \times \lambda_1 = 4.05 \times 10^{-3}$. Assuming uniformity in the flow rate throughout the entire system MC, the determined flow rate signifies the flow rate of the entire system. Lastly, utilizing the equation for calculating the

**Table 3. Operation time of each transition in ARIFT for incoming operation process.**

| Transition | t1 | t2 | t3 | t4 | t5 | t6 | t7 | t8 | t9 | t10 | t11 | t12 |
|---|---|---|---|---|---|---|---|---|---|---|---|---|
| Time/min | 10 | 5 | 5 | 10 | 5 | 20 | 150 | 30 | 5 | 30 | 80 | 30 |
| Transition | t13 | t14 | t15 | t16 | t17 | t18 | t19 | t20 | t21 | t22 | t23 | t24 |
| Time/min | 40 | 80 | 60 | 15 | 60 | 20 | 20 | 70 | 10 | 10 | 15 | 12 |

**Table 4. Operational time and implementation rate for each transition in ARIFT.**

| Transition | min | Rate | 1/min | Transition | min | Rate | 1/min |
|---|---|---|---|---|---|---|---|
| t1 | 10 | $\lambda_1$ | 0.1 | t14 | 80 | $\lambda_{14}$ | 0.0125 |
| t2 | 5 | $\lambda_2$ | 0.2 | t15 | 60 | $\lambda_{15}$ | 0.017 |
| t3 | 5 | $\lambda_3$ | 0.2 | t16 | 15 | $\lambda_{16}$ | 0.067 |
| t4 | 10 | $\lambda_4$ | 0.083 | t17 | 60 | $\lambda_{17}$ | 0.017 |
| t5 | 5 | $\lambda_5$ | 0.2 | t18 | 20 | $\lambda_{18}$ | 0.05 |
| t6 | 20 | $\lambda_6$ | 0.05 | t19 | 20 | $\lambda_{19}$ | 0.05 |
| t7 | 150 | $\lambda_7$ | 0.007 | t20 | 70 | $\lambda_{20}$ | 0.014 |
| t8 | 30 | $\lambda_8$ | 0.033 | t21 | 10 | $\lambda_{21}$ | 0.1 |
| t9 | 5 | $\lambda_9$ | 0.2 | t22 | 10 | $\lambda_{22}$ | 0.1 |
| t10 | 30 | $\lambda_{10}$ | 0.033 | t23 | 15 | $\lambda_{23}$ | 0.067 |
| t11 | 80 | $\lambda_{11}$ | 0.0125 | t24 | 12 | $\lambda_{24}$ | 0.083 |
| t12 | 30 | $\lambda_{12}$ | 0.033 | t25 | 2 | $\lambda_{25}$ | 0.5 |
| t13 | 40 | $\lambda_{13}$ | 0.025 | | | | |

average delay of the system, the average delay of the entire incoming freight process system is computed as $N = \bar{N}/\lambda = 304.272$ minutes.

**Optimization strategies for ARIFT incoming operation processes.** The previous section conducted an analysis of the PN model representing the domestic ARIFT incoming freight process, then established the MC to calculate the occupancy rates of each stage within the system and the overall system delay time. Subsequently, based on the performance analysis results, the ECRS principles and the principles of reorganization and optimization of the adjacency matrix (refer to Tables 8 and 9), this section proceeds to optimize the model.

It is observed that t20 in Table 2, being part of a synchronous relationship, can potentially affect the efficiency of the entire process. Therefore, p22 is deleted to directly connect t19 with p20, thereby transforming the synchronous relationship of t20 into a concurrent relationship that preserves the network structure.

The analysis of the occupancy rates of places based on the performance analysis results (as shown in Table 7) reveals that p3, p4, p13 and p16 have relatively high occupancy rates. High-occupancy places within the main process are known to impact the efficiency of the entire system. Among these, p13, being part of a branching process, does not significantly affect the efficiency of the entire system. However, p3, p4, and p16, as integral components of the main process, require optimization based on ECRS principles. The optimization strategy encompasses the following four aspects:

1. Optimization of Order Verification Process after Aircraft Arrival

Errors sometimes occur during the order verification process, leading to a loop structure in the original model. The preparation status for order verification corresponds to p4. Due to the

**Table 5. Marking states and included places in PN for incoming operation in ARIFT.**

| Marking | M0 | M1 | M2 | M3 | M4 | M5 | M6 | M7 |
|---|---|---|---|---|---|---|---|---|
| Place | p1 | p2 | p3 | p4 | p5 | p6 | p7 | p8 |
| Marking | M8 | M9 | M10 | M11 | M12 | M13 | M14 | M15 |
| Place | p9/p13 | p10/p13 | p11/p13 | p12/p13 | p9/p14 | p10/p14 | p11/p14 | p12/p14 |
| Marking | M16 | M17 | M18 | M19 | M20 | M21 | M22 | M23 |
| Place | p15 | p16 | p17 | p18/p21 | p19/p21 | p20/p21 | p18/p22 | p19/p22 |
| Marking | M24 | M25 | M26 | M27 | M28 | M29 | | |
| Place | p20/p22 | p23 | p24 | p25 | p26 | p27 | | |

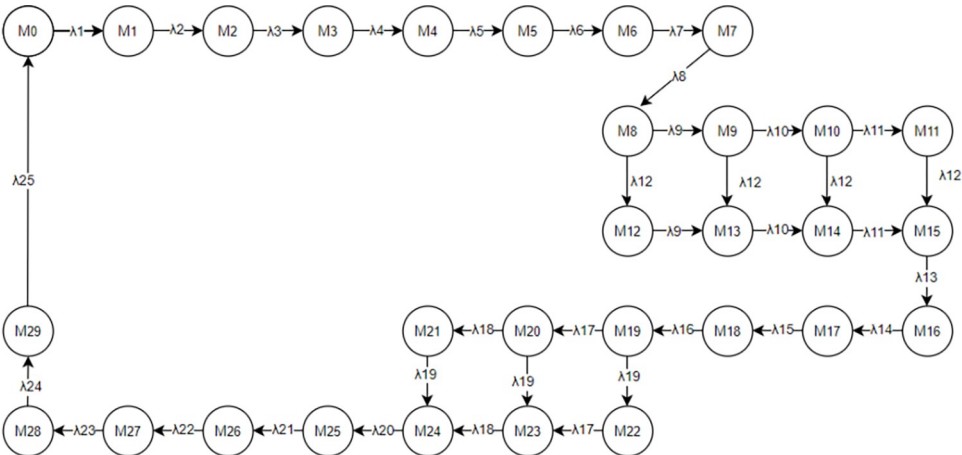

**Fig 4. MC of incoming operation process in ARIFT.**

high occupancy rate of p4, simplifying the loop structure in the model is necessary. This involves removing the steps for verifying and handling erroneous orders, merging them with the order verification step. To maintain the rationality after deletion, the optimized order verification step will be extended by 2 minutes.

2. Optimization of the Unloading Process After Aircraft Arrival

Because of the high occupancy rate of unloading preparation personnel (p3), it is necessary to parallel this process with the main process to enhance overall system efficiency. According to the ECRS principle, the arrival of unloading personnel and the formulation of unloading plans can be synchronized with the main process steps. The model will parallel the steps of unloading personnel arrival and plan formulation with the steps of aircraft entering the apron and order verification, forming a selection structure. This reduces the occupancy rate of places in the main process and minimizes overall system delay.

3. Optimization of Railway Container Installation Process

Due to the high occupancy rate at p16 where rail intermodal containers are prepared for installation, it needs to be parallel to the main process. Based on the ECRS principle, it can be inferred that there is no strict sequential relationship between the container installation and the cargo assembly and weighing processes, demonstrating that this step can be parallelized with other steps. Taking into account the comparable durations of rail intermodal container installation and disassembly of air freight containers, and referring to Table 7 which indicates

**Table 6. Marking probability of PN for ARIFT incoming operation.**

| Marking | M0 | M1 | M2 | M3 | M4 | M5 | M6 | M7 |
|---|---|---|---|---|---|---|---|---|
| Probability | 0.04050 | 0.00096 | 0.10008 | 0.10031 | 0.00103 | 0.00118 | 0.00194 | 0.00076 |
| Marking | M8 | M9 | M10 | M11 | M12 | M13 | M14 | M15 |
| Probability | 0.08780 | 0.O0869 | 0.08694 | 0.00167 | 0.00075 | 0.00246 | 0.00098 | 0.00577 |
| Marking | M16 | M17 | M18 | M19 | M20 | M21 | M22 | M23 |
| Probability | 0.00443 | 0.11916 | 0.01302 | 0.09188 | 0.00699 | 0.00097 | 0.00024 | 0.00031 |
| Marking | M24 | M25 | M26 | M27 | M28 | M29 | | |
| Probability | 0.00369 | 0.10542 | 0.01458 | 0.00150 | 0.09920 | 0.00773 | | |

**Table 7. Occupancy rate of places in PN for ARIFT incoming operation.**

| Place | Occupancy rate | Place | Occupancy rate | Place | Occupancy rate | Place | Occupancy rate |
|---|---|---|---|---|---|---|---|
| p1 | 0.04050 | p9 | 0.08863 | p17 | 0.01302 | p25 | 0.00150 |
| p2 | 0.00096 | p10 | 0.03329 | p18 | 0.09212 | p26 | 0.09920 |
| p3 | 0.10008 | p11 | 0.08792 | p19 | 0.00730 | p27 | 0.00773 |
| p4 | 0.10031 | p12 | 0.00744 | p20 | 0.00466 | | |
| p5 | 0.00103 | p13 | 0.18510 | p21 | 0.09984 | | |
| p6 | 0.00118 | p14 | 0.00996 | p22 | 0.00424 | | |
| p7 | 0.00194 | p15 | 0.00443 | p23 | 0.10542 | | |
| p8 | 0.00076 | p16 | 0.11916 | p24 | 0.01458 | | |

a low occupancy rate at p8 before disassembly of air freight containers, it is feasible to introduce a process branch. Therefore, the intermodal container installation process on the rail can be paralleled with the processes of disassembling air cargo containers and transporting them to the belly hold.

4. Optimization of the Document Handover Process before High-Speed Train Departure

Following the ECRS principle, the document handover process can be parallelized with the main process. Document handover can occur simultaneously with the loading of goods, and according to Table 7, the occupancy rate at p20 where goods are prepared for loading is relatively low, suggesting the capability to introduce process branches. Hence, the document handover process can be parallelized with the goods loading process.

## Optimization PN model and performance analysis of ARIFT incoming operation processes

**Optimized PN of incoming operations processes in ARIFT.** The model underwent analysis and optimization based on performance analysis using MC, ECRS principles, and the principles of reorganization and optimization of adjacency matrix. Optimization mainly targeted four aspects of the original model: order verification, unloading, packaging of railway containers, and document handover. This aimed to decrease the busy rate of the main process locations and diminish overall system latency through parallel processes. The optimized PN along with the meanings of places and transitions is depicted in Fig 5 and Table 10.

**Performance analysis of optimized incoming operation process in ARIFT.** With the removal of the "processing erroneous orders" step in the optimized model, one transition has been eliminated, while the data for other transitions remain largely unchanged. The timings for the 23 transitions after optimization are displayed in Table 11, while the implementation rates of these transitions are depicted in Table 12. Constructing the reachable marking graph

**Table 8. ECRS principles.**

| Principle | Meaning |
|---|---|
| Cancellation | Attempt to eliminate non-value-added or ineffective activities from existing processes; for processes that are difficult to eliminate, efforts should be made to avoid or reduce their occurrences. |
| Consolidation | Merge processes with operations or organizations from other processes based on practical circumstances. |
| Rearrangement | Analyze the sequence of steps in each process and make adjustments accordingly to reduce redundancy and enhance operational efficiency. |
| Simplification | Simplify organizational structures, operations, and actions involved in the process. |

Table 9. Principles of reorganization and optimization of the adjacency matrix.

| Principle | Meaning |
|---|---|
| Consolidation | When conflicts arise between transitions, processes can be merged to eliminate resource contention. |
| Deletion | Processes with choice relationships may lead to idle resources. Removing unnecessary transitions while retaining those with triggers can mitigate this issue. |
| Gap Reduction | Transitions and activities create interdependencies. Processes with dependencies need to minimize the time gap between them before synchronization to enhance overall efficiency. |
| Retention | Concurrent transitions with no resource occupation or idle situation can be retained. |

based on the optimized PN reveals the relationship between markings and places, depicted in Table 13. Assigning values to the reachable marking graph based on known transition implementation rates yields an isomorphic MC corresponding to the optimized PN, depicted in Fig 6.

The optimized MC can generate the corresponding transition rate matrix Q'. Assuming the probability of all markings in the optimized MC is $X' = (x_1, x_2, x_3, \ldots, x_{32}, x_{33})$, the steady-state probabilities of markings in the optimized MC can be obtained according to Eq (1), as shown in Table 14. Furthermore, utilizing the relationship between steady-state probabilities of markings and places (as shown in Table 13) along with Eq (2), the busy rates of places after optimization can be derived and are presented in Table 15.

The average token count of places in the optimized PN is correlated with the busy rates of places. Therefore, the average token count of the system is $\bar{N} = \sum_{i=1}^{26} u_i = \sum_{i=1}^{26} [p(Mi) = 1] = 1.59541$. The inflow rate of tokens into the system is $u_1 \times \lambda_1 = 5.68 \times 10^{-3}$. As the flow rates of the MC in the optimized model are consistent, the calculated flow rate represents the overall system flow rate. Finally, using the equation for average system latency, the average latency of the incoming operation process system after optimization is calculated to be $N = \bar{N}/\lambda = 280.882$ minutes.

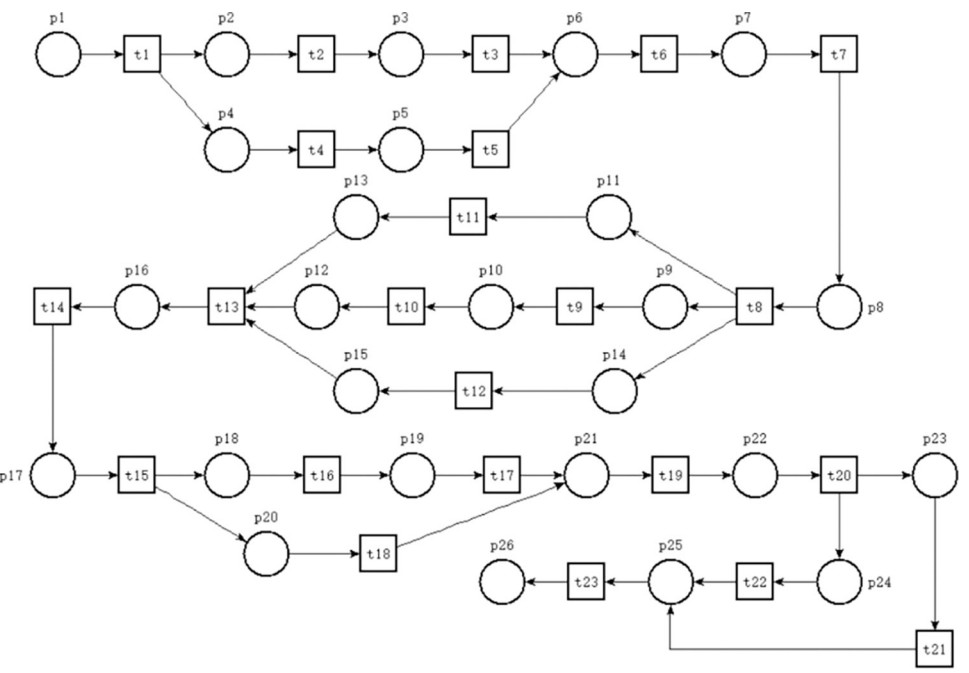

Fig 5. PN of optimized domestic inbound operations process in ARIFT.

**Table 10. Meanings of places and transitions in the optimized PN for domestic incoming operations in ARIFT.**

| Place (p) | Meaning | Transition (t) | Meaning |
|---|---|---|---|
| p1 | Aircraft arrives at the airport, prepares for landing | t1 | Aircraft is landing |
| p2 | Aircraft completes landing, prepares to enter the apron | t2 | Aircraft is entering the apron |
| p3 | Cargo orders are being prepared for verification | t3 | Orders are being verified |
| p4 | Freight personnel prepare to enter | t4 | Freight personnel are entering |
| p5 | Freight personnel prepare to develop unloading plan | t5 | Freight personnel are developing unloading plan |
| p6 | Freight personnel prepare for unloading | t6 | Freight personnel are unloading |
| p7 | Unloading completed, cargo ready for apron trailer transport | t7 | Cargo is being transported by apron trailer |
| p8 | Cargo arrives at the cargo handling area, prepares to confirm transportation mode | t8 | Confirming transportation mode |
| p9 | Cargo confirmed for all-cargo aircraft transportation | t9 | Cargo is being transported by all-cargo aircraft |
| p10 | Prepare for disassembly of air cargo containers | t10 | Air cargo containers are being disassembled |
| p11 | Cargo confirmed for belly transportation | t11 | Cargo is being transported via belly |
| p12 | Disassembly completed, prepare to transport cargo to sorting area | t12 | Packaging of railway containers |
| p13 | Belly transportation completed, cargo ready for transport to sorting area | t13 | Cargo is being transported to sorting area |
| p14 | Prepare for packaging of railway containers | t14 | Cargo is being assembled |
| p15 | Packaging of railway containers completed, cargo ready for transport to sorting area | t15 | Cargo is being transported to weighing area |
| p16 | Cargo is ready for reassembly | t16 | Cargo is being weighed |
| p17 | Cargo is about to arrive at the weighing area | t17 | Cargo is transported by apron trailer |
| p18 | Cargo is ready for weighing | t18 | Loading personnel are making loading plan |
| p19 | Cargo weighing completed, prepare for apron trailer transport | t19 | Loading personnel are loading |
| p21 | Cargo is ready for loading | t21 | Pre-departure inspection of cargo |
| p22 | Loading completed, preparing for post-loading inspection | t22 | Cargo documents are being handed over |
| p23 | Cargo is ready for pre-departure inspection | t23 | Cargo is leaving the air-rail intermodal center |
| p24 | Cargo is ready for document handover | | |
| p25 | High-speed rail is ready for departure | | |
| p26 | High-speed rail leaves the air-rail intermodal center, enters the network | | |

## Result and discussion

Performance analysis of the pre-optimized ARIFT incoming process PN model indicates that the high busy rate of departmental places in the original model affects the overall process delay. This paper optimizes the original model using the ECRS and correlation matrix reorganization optimization principles, resulting in an improved ARIFT incoming model. To verify the effectiveness of the optimization measures, we conducted a performance analysis of the optimized model using MC theory, and obtained the post-optimization performance results. A comparison of performance indicators for the incoming operation process before and after optimization is presented in Table 16.

The table shows that the average number of markings in the system increased from 1.23230 in the original model to 1.59541 in the optimized model. The 29.5% increase is because the optimized model's MC has more branches and markings, making the relationship between

**Table 11. Operation time for each link in the optimized incoming ARIFT.**

| Transition | t1 | t2 | t3 | t4 | t5 | t6 | t7 | t8 | t9 | t10 | t11 | t12 |
|---|---|---|---|---|---|---|---|---|---|---|---|---|
| Time/min | 10 | 5 | 15 | 5 | 20 | 150 | 30 | 5 | 30 | 80 | 30 | 60 |
| Transition | t13 | t14 | t15 | t16 | t17 | t18 | t19 | t20 | t21 | t22 | t23 | |
| Time/min | 40 | 80 | 15 | 60 | 20 | 20 | 70 | 10 | 15 | 10 | 12 | |

**Table 12. Operation time and implementation rate for each link in the optimized incoming ARIFT.**

| Transition | min | Rate | 1/min | Transition | min | Rate | 1/min |
|---|---|---|---|---|---|---|---|
| t1 | 10 | $\lambda_1$ | 0.1 | t14 | 80 | $\lambda_{14}$ | 0.0125 |
| t2 | 5 | $\lambda_2$ | 0.2 | t15 | 15 | $\lambda_{15}$ | 0.067 |
| t3 | 15 | $\lambda_3$ | 0.067 | t16 | 60 | $\lambda_{16}$ | 0.0167 |
| t4 | 5 | $\lambda_4$ | 0.2 | t17 | 20 | $\lambda_{17}$ | 0.05 |
| t5 | 20 | $\lambda_5$ | 0.05 | t18 | 20 | $\lambda_{18}$ | 0.05 |
| t6 | 150 | $\lambda_6$ | 0.0067 | t19 | 70 | $\lambda_{19}$ | 0.0143 |
| t7 | 30 | $\lambda_7$ | 0.033 | t20 | 10 | $\lambda_{20}$ | 0.1 |
| t8 | 5 | $\lambda_8$ | 0.2 | t21 | 15 | $\lambda_{21}$ | 0.067 |
| t9 | 30 | $\lambda_9$ | 0.033 | t22 | 10 | $\lambda_{22}$ | 0.1 |
| t10 | 80 | $\lambda_{10}$ | 0.0125 | t23 | 12 | $\lambda_{23}$ | 0.083 |
| t11 | 30 | $\lambda_{11}$ | 0.033 | t24 | 2 | $\lambda_{24}$ | 0.5 |
| t12 | 60 | $\lambda_{12}$ | 0.0167 | | | | |
| t13 | 40 | $\lambda_{13}$ | 0.025 | | | | |

**Table 13. Marking states and included places in PN for incoming operation ARIFT.**

| Marking | M0 | M1 | M2 | M3 | M4 | M5 | M6 |
|---|---|---|---|---|---|---|---|
| Probability | p1 | p2/p4 | p2/p5 | p3/p4 | p3/p5 | p6/p4 | p6/p5 |
| Marking | M7 | M8 | M9 | M10 | M11 | M12 | M13 |
| Probability | p6 | p7 | p8 | p9/p11 /p14 | p9/p11 /p15 | p9/p13 /p14 | p9/p13 /p15 |
| Marking | M14 | M15 | M16 | M17 | M18 | M19 | M20 |
| Probability | p10/p11 /p14 | p10/p11 /p15 | p10/p13 /p14 | p10/p13 /p15 | p12/p11 /p14 | p12/p11 /p15 | p12/p13 /p14 |
| Marking | M21 | M22 | M23 | M14 | M25 | M26 | M27 |
| Probability | p12/p13 /15 | p16 | p17 | p18/p20 | p19/p20 | p21/p20 | p21 |
| Marking | M28 | M29 | M30 | M31 | M32 | | |
| Probability | p22 | p23/p24 | p25/p24 | p25 | p26 | | |

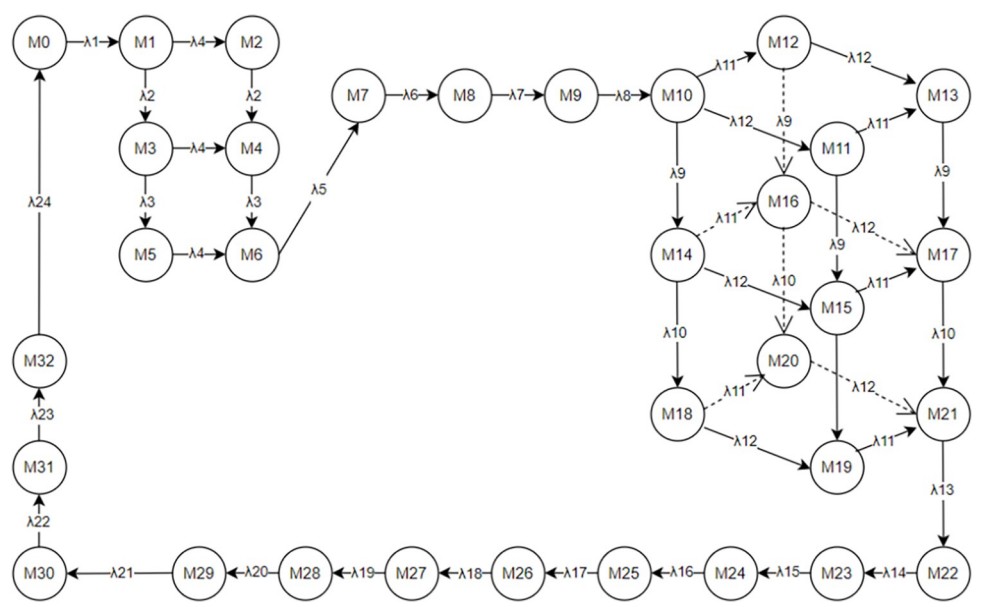

**Fig 6. MC of the optimized incoming ARIFT.**

**Table 14. Marking probability in PN for optimized incoming ARIFT.**

| Marking | M0 | M1 | M2 | M3 | M4 | M5 | M6 |
|---|---|---|---|---|---|---|---|
| Probability | 0.05680 | 0.00085 | 0.00063 | 0.00007 | 0.00004 | 0.00101 | 0.00221 |
| Marking | M7 | M8 | M9 | M10 | M11 | M12 | M13 |
| Probability | 0.00244 | 0.16905 | 0.02551 | 0.02544 | 0.01640 | 0.04860 | 0.04330 |
| Marking | M14 | M15 | M16 | M17 | M18 | M19 | M20 |
| Probability | 0.00291 | 0.00008 | 0.02747 | 0.01723 | 0.00236 | 0.00820 | 0.00663 |
| Marking | M21 | M22 | M23 | M24 | M25 | M26 | M27 |
| Probability | 0.02913 | 0.06445 | 0.14941 | 0.00251 | 0.02784 | 0.05869 | 0.07445 |
| Marking | M28 | M29 | M30 | M31 | M32 | | |
| Probability | 0.05842 | 0.00983 | 0.03764 | 0.02032 | 0.01015 | | |

**Table 15. Place occupancy rate in PN for optimized incoming ARIFT.**

| Place | Occupancy rate | Place | Occupancy rate | Place | Occupancy rate | Place | Occupancy rate |
|---|---|---|---|---|---|---|---|
| p1 | 0.05680 | p9 | 0.13374 | p17 | 0.14941 | p25 | 0.05796 |
| p2 | 0.00148 | p10 | 0.04769 | p18 | 0.00251 | p26 | 0.01015 |
| p3 | 0.00011 | p11 | 0.05539 | p19 | 0.02784 | | |
| p4 | 0.00108 | p12 | 0.04632 | p20 | 0.08904 | | |
| p5 | 0.00225 | p13 | 0.17236 | p21 | 0.13314 | | |
| p6 | 0.00566 | p14 | 0.11341 | p22 | 0.05842 | | |
| p7 | 0.16905 | p15 | 0.11434 | p23 | 0.00983 | | |
| p8 | 0.02551 | p16 | 0.06445 | p24 | 0.04747 | | |

**Table 16. Comparison of performance indicator of incoming operation before and after optimization.**

| Performance Indicator | Before Optimization | After Optimization | Change Rate |
|---|---|---|---|
| Average marking count | 1.23230 | 1.59541 | 29.5% |
| Average marking flow rate | $4.05 \times 10^{-3}$ | $5.68 \times 10^{-3}$ | 40.2% |
| Average delay time/min. | 304.272 | 280.882 | -7.7% |

places and markings more complex. Additionally, the average marking flow rate increased from $4.05 \times 10^{-3}$ in the original model to $5.68 \times 10^{-3}$ in the optimized model, representing a 40.2% increase. This increase occurred because the main process's marking flow rate remained unchanged, but some segments were converted from a single to a parallel structure to reduce total system delay. Fig 2 shows the original model with two sets of parallel structures, each generating two branches. Fig 5 shows the optimized model with four sets of parallel structures, with one set generating three branches. The increase in branches leads to a higher average marking flow rate, making the 40.2% increase reasonable. The table shows a 29.5% increase in the average count of markings and a 40.2% increase in the average marking flow rate. This indicates that the increase in the average number of markings is less than the increase in the average marking flow rate. Based on the equation for system average delay ($N = \bar{N}/\lambda$), the total delay time of the system is reduced after optimization. Therefore, the optimization of the ARIFT incoming process using SPN and MC research methods is feasible. This implies that SPN and MC research methods can be applied to analyze more operational processes, as long as all transitions have delays and there are no instantaneous transitions. Furthermore, in studying ARIT operational processes, we can use partial simulation tools and perform mathematical analysis using SPN and MC methods. Compared to simulation tools, SPN and MC

methods calculate the total delay and provide insights into changes in each model segment by calculating the place occupancy rate. These insights better inform data-driven optimization measures, making them more objective. In summary, this highlights the advantages of using SPN and MC theories to optimize ARIT operational processes.

However, research on ARIT operational processes will not be limited to this. This paper focuses on the incoming operational process of ARIT cargo transport, while ARIT includes both incoming and outbound operations. Therefore, future research will focus on the outbound operational process of ARIT freight transport. And future studies will consider using more complex Petri net tools for process modeling and analysis. This study only incorporates time data as a reference for ARIT research and does not include informational content within the operational processes. This omission prevents specific analyses on individual cargo. Tables 3 and 11 show that these reference data are fixed and non-zero values, as SPN and MC performance analysis can only be conducted based on fixed and non-zero data. This is a limitation because, in practical freight operations, process running time generally fluctuates within a certain range. Future optimization studies of fluctuating operational processes will employ colored Petri net (CPN) tools, incorporating more time and information flows into the model.

## Conclusion

This paper comprehensively studies the ARIFT incoming operations process in China. First, we investigate Zhengzhou Xinzheng International Airport, Zhengzhou South Station, and the air-rail intermodal freight center between them. We analyze and summarize the specific activities, especially the freight operations of these three departments, and derive the ARIFT incoming operations workflow diagram. Based on PN theory, we transform the ARIFT incoming workflow diagram into a corresponding PN model. Using SPN and MC research methods, we analyze the relationship between places and markings and construct the corresponding MC for the PN model. By constructing the MC, we calculate the steady-state probability of each marking and use these probabilities to determine the place occupancy rate and total delay of the PN model. The place occupancy rate is a key metric for analyzing process step efficiency in the model, helping identify steps that are relatively cumbersome and need optimization. Our calculations reveal that the busy rates of places p3, p4, p13, and p16 are particularly high, indicating surrounding process steps require optimization. By applying the ECRS principle, we optimize the model and derive a new PN model for ARIFT incoming operations. Finally, using the aforementioned performance analysis methods, we calculate the total delay of the optimized model. The following are some conclusions from this study:

1. Based on PN theory, we modeled the ARIFT incoming operations process. By integrating the properties of SPN and transition timings, we constructed the model's MC. Using MC theory, we conducted a performance analysis of the model. The performance analysis results allow for intuitive identification of process steps that require optimization.

2. Using the model performance analysis results, the ECRS principle, and the association matrix reorganization optimization principle, we optimized certain parts of the model. Based on SPN theory, we constructed the MC for the optimized model and performed a performance analysis. The analysis revealed that the original model's average system delay was 304.272 minutes, while the optimized model's average system delay was reduced to 280.882 minutes, a 7.7% decrease. This demonstrates the feasibility of the model optimization.

3. In the ARIFT process, the loop structure of the order review stage has been transformed into a continuous structure. This implies that order review accuracy needs to be improved.

More precise review systems, such as those using computers or artificial intelligence, should be implemented to reduce manual review errors and minimize time wasted due to review errors.

4. In the original ARIFT process, the aircraft landing and unloading plan formulation were conducted sequentially. After optimization, unloading personnel need to be pre-informed about the cargo distribution and configuration in the cabin. This allows the unloading plan formulation to occur simultaneously with the aircraft landing, thereby reducing the time previously required for plan formulation.

5. In the original ARIFT process, the railway container packaging stage occurred after the cargo distribution stage, and the document handover stage took place sequentially with the cargo loading stage. Additionally, the document handover and cargo loading stages now occur concurrently. This optimization leverages departmental collaboration and information sharing, reducing the total delay in the ARIFT incoming process without affecting its continuity. The overall reduction in the total delay of the ARIFT incoming process significantly reduces the total time for air-rail intermodal freight transportation. This fully leverages the advantage of fast transportation in air-rail intermodal logistics, reducing time costs.

This paper has some limitations, necessitating further research on the ARIFT process. The following are some limitations and recommendations:

1. SPN and MC research methods can reasonably optimize the model based on data results. However, this study only considers time flow, which is relatively narrow. Future research should incorporate information flow into process optimization to ensure validity and practicality.

2. The MC performance analysis method only accounts for fixed time factors. More complex Petri net tools, such as CPN, should be used to study fluctuating time flow and information flow in ARIFT.

3. This study only examines the ARIFT incoming operations process. Future research should also focus on the ARIFT outbound operations process. Considering both import and export scenarios will provide a more comprehensive understanding of ARIFT operations.

## Author Contributions

**Data curation:** Yihu Lei.

**Formal analysis:** Yihu Lei.

**Investigation:** Yihu Lei.

**Software:** Yihu Lei.

**Supervision:** Haibo Mu.

**Validation:** Haibo Mu.

**Writing – original draft:** Yihu Lei.

**Writing – review & editing:** Yihu Lei, Haibo Mu.

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
