## [Decision Letter · Decision Letter 0]

8 May 2024

PONE-D-24-13474Analysis and Optimization of a Stochastic Petri Net for Air-Rail Intermodal TransportationPLOS ONE

Dear Dr. Lei,

Thank you for submitting your manuscript to PLOS ONE. After careful consideration, we feel that it has merit but does not fully meet PLOS ONE’s publication criteria as it currently stands. Therefore, we invite you to submit a revised version of the manuscript that addresses the points raised during the review process.

We look forward to receiving your revised manuscript.

Kind regards,

Nhat-Luong Nhieu, Asst. Prof.

Academic Editor

PLOS ONE

Journal Requirements:

2. Please note that your Data Availability Statement is currently missing the repository name and/or the DOI/accession number of each dataset OR a direct link to access each database. If your manuscript is accepted for publication, you will be asked to provide these details on a very short timeline. We therefore suggest that you provide this information now, though we will not hold up the peer review process if you are unable.

Additional Editor Comments:

Based on the review reports, the authors need to make some edits and address the reviewer's concerns.

Reviewers' comments:

Reviewer's Responses to Questions

**Comments to the Author**

1. Is the manuscript technically sound, and do the data support the conclusions?

Reviewer #1: Yes

Reviewer #2: Yes

Reviewer #3: Partly

2. Has the statistical analysis been performed appropriately and rigorously? 

Reviewer #1: Yes

Reviewer #2: Yes

Reviewer #3: N/A

3. Have the authors made all data underlying the findings in their manuscript fully available?

Reviewer #1: Yes

Reviewer #2: Yes

Reviewer #3: No

4. Is the manuscript presented in an intelligible fashion and written in standard English?

Reviewer #1: Yes

Reviewer #2: Yes

Reviewer #3: Yes

5. Review Comments to the Author

Reviewer #1: - The parentheses and texts need the space. For example: "Air-rail intermodal transportation (ARIT)". Please revise them in all points of the paper, including following format of the journal template.

- In the beginning of introduction part, before mentioning about intermodal transportation in China, 1-2 sentences discussing about general informantion on intermodal transportation in global view with citations are required.

Reviewer #2: 1. the abstract is not clear enough, add aim, scope, novelty and last the main obtained results

2. need more discussions for outcomes of research.

3. conclusions are weak, need to improve

4. add recommendation and limitation of the work.

Reviewer #3: Please extend on the mathematical modelling of the problem, restrictions, and interpretation of results. The complexity of the problem is not explained (what is the relevance or importance of solving this problem to optimality?)

6. PLOS authors have the option to publish the peer review history of their article (what does this mean?). If published, this will include your full peer review and any attached files.

Reviewer #1: **Yes: **Asst.Prof.Dr.Wissawa Aunyawong

Reviewer #2: No

Reviewer #3: No

---

## [Author Response · Author response to Decision Letter 0]

25 Jun 2024

Dear Editors，

I hope this message finds you well.

We are sincerely grateful for the valuable feedback and suggestions provided by you and the reviewers for our manuscript titled “[Analysis and optimization of a Stochastic Petri Net for Air-Rail Intermodal Transportation].” We have discussed and revised according to the valuable suggestions from the editors for a period of time, and have made efforts to meet their requirements. Regarding the issue of the Data Availability Statement, since the data in the article is based on the data from reference [13], I have inserted the DOI of reference [13] into the references. In addition, we have thoroughly reviewed all the comments and have made comprehensive revisions to enhance the quality and rigor of our paper.

Attached, please find the revised version of our manuscript along with a detailed response to the reviewers’ comments. In the response document, we have addressed each comment individually and explained the specific changes made. Below is a summary of the major revisions:

Reviewer #1 Comments:

Comment 1: [The parentheses and texts need the space. For example: "Air-rail intermodal transportation (ARIT)". Please revise them in all points of the paper, including following format of the journal template.]

Response: Thank you for your firstly valuable feedback. I have revised the manuscript according to your suggestions. Specifically, I have ensured that there is a space before parentheses throughout the paper, such as in "Air-rail intermodal transportation (ARIT)". Additionally, I have reviewed the entire document to ensure it adheres to the formatting guidelines of the journal template.

Comment 2: [In the beginning of introduction part, before mentioning about intermodal transportation in China, 1-2 sentences discussing about general informantion on intermodal transportation in global view with citations are required.]

Response: Thank you for your secondly insightful feedback. I have added 1-2 sentences at the beginning of the introduction to provide general information on intermodal transportation from a global perspective, along with appropriate citations. This addition helps to set the context before discussing intermodal transportation in China.

The revised sentences are as follows:

Significant progress has been made in the development of global Intermodal transportation all around the world, especially in Europe and China. Europe has gained extensive experience from early research and practice, significantly enhancing operational efficiency and passenger convenience through system optimization and intermodal cooperation.

Thank you for your valuable feedback and the two insightful comments. I have addressed each point and made the necessary revisions accordingly. If there are any further adjustments required, please let me know.

Reviewer #2 Comments:

Comment 1: [the abstract is not clear enough, add aim, scope, novelty and last the main obtained results]

Response: Thank you for your firstly valuable feedback on the abstract. I have revised the abstract to include the aim, scope, novelty, and main obtained results as suggested. The revised abstract is as follows:

Abstract

Air-rail intermodal transportation (ARIT) plays a crucial role in China's intermodal transportation system. This study aims to model and optimize issues such as inefficiency and complexity in China's ARIT freight transportation using Business Process Reengineering and Stochastic Petri Nets theories. The Petri Net (PN) model for incoming freight transport in ARIT is based on actual operations, employing a new method involving Stochastic Petri Nets and isomorphic Markov Chains theory for performance analysis. Performance analysis results help intuitively identify areas needing optimization. Based on optimization principles, elements such as railway container packaging are improved, resulting in an optimized PN model for ARIT. Finally, data analysis shows that the optimized ARIT model reduces total delay by 7.7% compared to the original. This demonstrates that the new method, combining Markov Chain performance analysis and optimization principles, is feasible and effective for ARIT optimization.

Comment 2: [need more discussions for outcomes of research]

Response: Thank you for your feedback on the outcomes of the research. I have expanded the discussion of the research outcomes to provide a more detailed explanation of our findings and their significance. According to the journal's standards, I have added a "Results and Discussion" section, providing a brief description of the previous experiments. Additionally, I have included a comparative table of performance before and after the analysis. The content of the table is thoroughly analyzed and discussed, and the rationale behind the observed data is explained. Furthermore, I have added a final paragraph discussing the recommendations and limitations of the study, incorporating them into the overall discussion. The revised content is as follows:

Result and discussion

Performance analysis of the pre-optimized ARIFT incoming process PN model indicates that the high busy rate of departmental places in the original model affects the overall process delay. This paper optimizes the original model using the ECRS and correlation matrix reorganization optimization principles, resulting in an improved ARIFT incoming model. To verify the effectiveness of the optimization measures, we conducted a performance analysis of the optimized model using MC theory, and obtained the post-optimization performance results. A comparison of performance indicators for the incoming operation process before and after optimization is presented in Table 16.

Table 16. Comparison of Performance Indicator of Incoming Operation Before and After Optimization

Performance Indicator Before Optimization After Optimization Change Rate

Average marking count 1.23230 1.59541 29.5%

Average marking flow rate 4.0510-3 10-3 40.2%

Average delay time/min. 304.272 280.882 -7.7%

The table shows that the average number of markings in the system increased from 1.23230 in the original model to 1.59541 in the optimized model. The 29.5% increase is because the optimized model's MC has more branches and markings, making the relationship between places and markings more complex. Additionally, the average marking flow rate increased from 4.0510-3 in the original model to 10-3 in the optimized model, representing a 40.2% increase. This increase occurred because the main process's marking flow rate remained unchanged, but some segments were converted from a single to a parallel structure to reduce total system delay. Fig 2 shows the original model with two sets of parallel structures, each generating two branches. Fig 5 shows the optimized model with four sets of parallel structures, with one set generating three branches. The increase in branches leads to a higher average marking flow rate, making the 40.2% increase reasonable. The table shows a 29.5% increase in the average count of markings and a 40.2% increase in the average marking flow rate. This indicates that the increase in the average number of markings is less than the increase in the average marking flow rate. Based on the equation for system average delay(=/λ), the total delay time of the system is reduced after optimization. Therefore, the optimization of the ARIFT incoming process using SPN and MC research methods is feasible. This implies that SPN and MC research methods can be applied to analyze more operational processes, as long as all transitions have delays and there are no instantaneous transitions. Furthermore, in studying ARIT operational processes, we can use partial simulation tools and perform mathematical analysis using SPN and MC methods. Compared to simulation tools, SPN and MC methods calculate the total delay and provide insights into changes in each model segment by calculating the place occupancy rate. These insights better inform data-driven optimization measures, making them more objective. In summary, this highlights the advantages of using SPN and MC theories to optimize ARIT operational processes.

However, research on ARIT operational processes will not be limited to this. This paper focuses on the incoming operational process of ARIT cargo transport, while ARIT includes both incoming and outbound operations. Therefore, future research will focus on the outbound operational process of ARIT freight transport. And future studies will consider using more complex Petri net tools for process modeling and analysis. This study only incorporates time data as a reference for ARIT research and does not include informational content within the operational processes. This omission prevents specific analyses on individual cargo. Tables 3 and 11 show that these reference data are fixed and non-zero values, as SPN and MC performance analysis can only be conducted based on fixed and non-zero data. This is a limitation because, in practical freight operations, process running time generally fluctuates within a certain range. Future optimization studies of fluctuating operational processes will employ colored Petri net (CPN) tools, incorporating more time and information flows into the model.

Comment 3: [conclusions are weak, need to improve]

Response: Thank you for your feedback on the conclusions section. I have revised the conclusions to strengthen the content and logic. I have added a more detailed summary, providing a conclusive overview of the entire study. The optimized conclusions reflect both the overall and detailed findings, emphasizing the main discoveries and their significance. I have also discussed the potential applications of these findings. Furthermore, I have included some recommendations and limitations at the end of the conclusions to make it more comprehensive. The revised content is as follows: 

Conclusion

This paper comprehensively studies the ARIFT incoming operations process in China. First, we investigate Zhengzhou Xinzheng International Airport, Zhengzhou South Station, and the air-rail intermodal freight center between them. We analyze and summarize the specific activities, especially the freight operations of these three departments, and derive the ARIFT incoming operations workflow diagram. Based on PN theory, we transform the ARIFT incoming workflow diagram into a corresponding PN model. Using SPN and MC research methods, we analyze the relationship between places and markings and construct the corresponding MC for the PN model. By constructing the MC, we calculate the steady-state probability of each marking and use these probabilities to determine the place occupancy rate and total delay of the PN model. The place occupancy rate is a key metric for analyzing process step efficiency in the model, helping identify steps that are relatively cumbersome and need optimization. Our calculations reveal that the busy rates of places p3, p4, p13, and p16 are particularly high, indicating surrounding process steps require optimization. By applying the ECRS principle, we optimize the model and derive a new PN model for ARIFT incoming operations. Finally, using the aforementioned performance analysis methods, we calculate the total delay of the optimized model. The following are some conclusions from this study:

(1)Based on PN theory, we modeled the ARIFT incoming operations process. By integrating the properties of SPN and transition timings, we constructed the model's MC. Using MC theory, we conducted a performance analysis of the model. The performance analysis results allow for intuitive identification of process steps that require optimization.

(2)Using the model performance analysis results, the ECRS principle, and the association matrix reorganization optimization principle, we optimized certain parts of the model. Based on SPN theory, we constructed the MC for the optimized model and performed a performance analysis. The analysis revealed that the original model's average system delay was 304.272 minutes, while the optimized model's average system delay was reduced to 280.882 minutes, a 7.7% decrease. This demonstrates the feasibility of the model optimization.

(3)In the ARIFT process, the loop structure of the order review stage has been transformed into a continuous structure. This implies that order review accuracy needs to be improved. More precise review systems, such as those using computers or artificial intelligence, should be implemented to reduce manual review errors and minimize time wasted due to review errors.

(4)In the original ARIFT process, the aircraft landing and unloading plan formulation were conducted sequentially. After optimization, unloading personnel need to be pre-informed about the cargo distribution and configuration in the cabin. This allows the unloading plan formulation to occur simultaneously with the aircraft landing, thereby reducing the time previously required for plan formulation. 

(5)In the original ARIFT process, the railway container packaging stage occurred after the cargo distribution stage, and the document handover stage took place sequentially with the cargo loading stage. Additionally, the document handover and cargo loading stages now occur concurrently. This optimization leverages departmental collaboration and information sharing, reducing the total delay in the ARIFT incoming process without affecting its continuity. The overall reduction in the total delay of the ARIFT incoming process significantly reduces the total time for air-rail intermodal freight transportation. This fully leverages the advantage of fast transportation in air-rail intermodal logistics, reducing time costs.

This paper has some limitations, necessitating further research on the ARIFT process. The following are some limitations and recommendations:

(1)SPN and MC research methods can reasonably optimize the model based on data results. However, this study only considers time flow, which is relatively narrow. Future research should incorporate information flow into process optimization to ensure validity and practicality.

(2)The MC performance analysis method only accounts for fixed time factors. More complex Petri net tools, such as CPN, should be used to study fluctuating time flow and information flow in ARIFT. 

(3)This study only examines the ARIFT incoming operations process. Future research should also focus on the ARIFT outbound operations process. Considering both import and export scenarios will provide a more comprehensive understanding of ARIFT operations.

Comment 4: [add recommendation and limitation of the work.]

Response: Thank you for your feedback on the discussion and conclusion sections. I have revised these sections to strengthen their content and logic. I have added a more detailed summary, providing a conclusive overview of the entire study. The optimized conclusions reflect both the overall and detailed findings, emphasizing the main discoveries and their significance. I have also discussed the potential applications of these findings. According to your suggestion, I have included recommendations and limitations of the work at the end of the discussion and conclusion sections to make them more comprehensive. The revised content is as follows: 

limitations and recommendations(in Result and discussion)

However, research on ARIT operational processes will not be limited to this. This paper focuses on the incoming operational process of ARIT cargo transport, while ARIT includes both incoming and outbound operations. Therefore, future research will focus on the outbound operational process of ARIT freight transport. And future studies will consider using more complex Petri net tools for process modeling and analysis. This study only incorporates time data as a reference for ARIT research and does not include informational content within the operational processes. This omission prevents specific analyses on individual cargo. Tables 3 and 11 show that these reference data are fixed and non-zero values, as SPN and MC performance analysis can only be conducted based on fixed and non-zero data. This is a limitation because, in practical freight operations, process running time generally fluctuates within a certain range. Future optimization studies of fluctuating operational pr

---

## [Decision Letter · Decision Letter 1]

10 Jul 2024

Analysis and optimization of a Stochastic Petri Net for Air-Rail Intermodal Transportation

PONE-D-24-13474R1

Dear Dr. Lei,

We’re pleased to inform you that your manuscript has been judged scientifically suitable for publication and will be formally accepted for publication once it meets all outstanding technical requirements.

Kind regards,

Nhat-Luong Nhieu, Ph.D.

Academic Editor

PLOS ONE

Additional Editor Comments (optional): The review reports showed that the manuscript met the standards for publication.

Reviewers' comments:

Reviewer's Responses to Questions

**Comments to the Author**

1. If the authors have adequately addressed your comments raised in a previous round of review and you feel that this manuscript is now acceptable for publication, you may indicate that here to bypass the “Comments to the Author” section, enter your conflict of interest statement in the “Confidential to Editor” section, and submit your "Accept" recommendation.

Reviewer #1: All comments have been addressed

Reviewer #2: All comments have been addressed

2. Is the manuscript technically sound, and do the data support the conclusions?

Reviewer #1: Yes

Reviewer #2: Yes

3. Has the statistical analysis been performed appropriately and rigorously? 

Reviewer #1: Yes

Reviewer #2: Yes

4. Have the authors made all data underlying the findings in their manuscript fully available?

Reviewer #1: Yes

Reviewer #2: Yes

5. Is the manuscript presented in an intelligible fashion and written in standard English?

Reviewer #1: Yes

Reviewer #2: Yes

6. Review Comments to the Author

Reviewer #1: The comments in the introduction part and the other parts of the paper have been addressed. The paper now has been improved in terms o academic quality.

Reviewer #2: (No Response)

7. PLOS authors have the option to publish the peer review history of their article (what does this mean?). If published, this will include your full peer review and any attached files.

Reviewer #1: **Yes: **Asst.Prof.Dr.Wissawa Aunyawong

Reviewer #2: No

---

## [Editor Report · Acceptance letter]

12 Jul 2024

PONE-D-24-13474R1 

PLOS ONE

Dear Dr. Lei, 

I'm pleased to inform you that your manuscript has been deemed suitable for publication in PLOS ONE. Congratulations! Your manuscript is now being handed over to our production team.

Kind regards, 

on behalf of

Asst. Prof. Nhat-Luong Nhieu 

Academic Editor

PLOS ONE